# Application of Composite NanoMaterial to Determine Phenols in Wastewater by Solid Phase Micro Membrane Tip Extraction and Capillary Electrophoresis

**DOI:** 10.3390/molecules24193443

**Published:** 2019-09-23

**Authors:** Nora Hamad Al-Shaalan, Imran Ali, Zeid A. ALOthman, Lamya Hamad Al-Wahaibi, Hadeel Alabdulmonem

**Affiliations:** 1Department of Chemistry, P. O. Box 84428, College of Science, Princess Nourah bint Abdulrahman University, Riyadh 11671, Saudi Arabia; Lhalwahaib@pnu.edu.sa (L.H.A.-W.); haalabdulmenem@pnu.edu.sa (H.A.); 2Department of Chemistry, College of Sciences, Taibah University, Al-Medina Al-Munawara 41477, Saudi Arabia; 3Department of Chemistry, Jamia Millia Islamia, (Central University) New Delhi 11025, India; 4Department of Chemistry, P. O. Box 2455, College of Science, King Saud University, Riyadh 11451, Saudi Arabia; zaothman@ksu.edu.sa

**Keywords:** phenol, 3-aminophenol, wastewater, SPMMTE, capillary electrophoresis

## Abstract

Composite nanoparticles were used in solid phase micro membrane tip extraction and capillary electrophoresis to determine phenol and *p*-amino-phenol in wastewater. The optimized conditions were 100 g/L concentration, 40 min contact time, 11 pH, 5 mg/mL nanoparticles amounts, 60 min desorption time, 9 desorption pH and 298 K temperature. Capillary electrophoresis conditions were phosphate buffer (15 mM, pH 7.0) background electrolyte, 18 kV applied voltage, 214 nm UV detection, 30 s sample loading at 23 ± 1 °C. The maximum percent uptakes of *p*-amino-phenol and phenol were 80.0 and 85.0%. High ratio recoveries of *p*-amino-phenol and phenol from nanomaterial were 99.0 and 98. Consequently, the actual extractions of *p*-amino-phenol and phenol from wastewater were 79.2 and 83.30 percent. The migration times of phenol and *p*-amino-phenol and were 9.0 and 12.0 min. The detection limits of phenol and *p*-amino-phenols were 0.1 and 0.2 µg/L after extraction and CE. Therefore, this combination of solid phase micro membrane tip extraction and capillary electrophoresis may be considered as the ideal one for monitoring of toxic phenol and *p*-amino-phenol in water sample.

## 1. Introduction

The analysis of any sample of natural origin comprises two parts, that is, sample preparation and separation and identification. The first part of the experiment becomes important in case of the samples of biological and environmental origins. This is due to the fact that, sometimes, more than hundreds of impurities are present in the samples of the natural origin [1,2]. Hence, sample preparation is one of the crucial issues in the separation science. Many sample preparation methods are reported in the literature and these comprise liquid-liquid extraction, solid phase extraction, solid phase micro-extraction (SPME), supercritical fluid extraction, ultrasonic extraction, pressurized liquid extraction, empty fiber liquid phase micro-extraction, micro-wave assisted extraction, dispersive liquid-liquid micro-extraction, molecularly imprinted solid phase extraction, pressurized hot water extraction and solid phase micro membrane tip extraction (SPMMTE). Among these, the later method is supposed as the best one due to the involvement of low amount of adsorption media, fast extraction, capable to work at low concentration and low sample volume. This method is used for the extraction of some species from samples of biological and environmental importance [3,4]. Phenols are very notorious water pollutants and need to be monitored even at low concentration for which SPMMTE may be a choice of the sample preparation. Phenols have acute toxicities [5,6,7,8,9,10,11,12,13,14,15] along with tumorigenesis [16,17,18,19,20]. The collective causes of phenols pollution are industries related to dyes, textiles, pesticides, paper, pharmaceutical, tanning, plastic, gasoline, rubber and so forth [21,22]. Some systems are described for analyses but many of them consume high volume of toxic solvent and sorbent materials [23]. Besides, some techniques are not skilled for phenolic extraction at micro level concentration in water [24]. Taking these assessments into deliberation, composite iron nanomaterial is manufactured and used in SPMMTE for sample preparation of phenols containing water. Analysis of phenol was carried out by capillary electrophoresis [25].

## 2. Results and Discussion

The characterization of the synthesized composite iron nanoparticles, determination of phenols by capillary electrophoresis and optimization of the extraction of phenols in SPMMTE are discussed.

### 2.1. Characterization of the Synthesized Composite Iron NanoParticles

The prepared composite iron nanoparticles were categorized by XRD, TEM and SEM methods. The detail of the description is described elsewhere [26,27,28,29]. XRD peaks were observed at 2θ of 18.6 and 44.8; indicating presence of FeOOH and Fe^0^ (zero valent iron). Further XRD peak show amorphous nature of nanocomposite. TEM results reflect the roughness of the surface and SEM results indicates 10–30 nm particle size.

The supra molecular structure of the composite iron nanoparticle is revealed in Figure 1.

### 2.2. Determination of Phenols by Capillary Electrophoresis

Analysis of phenols in water was done by noting their migration times. The documentation of the separated phenols was carried out using standards of the phenols. The values of the migration times of phenol and 3-aminophenol in CE were 9.0 and 12.0 min. The capillary electrophoresis optimization was carried out by achieving the variation in the configuration of background electrolyte, pH of the background electrolyte, amount of sample loaded and capillary length and voltage. The best conditions were used in this paper. The validation data was also collected for this set of experiment and found ±0.70 to ±0.8 as standard deviation, 0.9997 to 0.9998 correlation coefficients and 98.8 to 99.0 confidence level.

### 2.3. Validation of Capillary Electrophoresis

The capillary electrophoretic method was validated as per standard procedure [30,31]. The valuables studied were linearity, specificity, limit of detection (LOD), limit of quantification (LOQ), accuracy, precision and ruggedness. The magnitudes of these variables are given in Table 1. It is lucid for the table that the developed method is accurate, specific, précised and rugged.

### 2.4. Extraction of Phenols by SPMMTE

To attain the maximal extraction of phenols with SPMMTE, numerous parameters were diverse and improved. The diverse settings were concentrations of phenols, pH, contact time, nanoparticles amount bounded in a cone of membrane and the temperature.

#### 2.4.1. Concentrations of Phenols

Primarily, with an augment in concentration from 25.0 to 100.0 µg L^−1^ the quantity of sorption in aminophenol and phenol progressively augmented from 5.0 to 16.0 μg/mg that is, 100–80%. Subsequently, more increase in concentration up to 150.0 µg L^−1^, the sorption of both remained persistent. Extra limitations; interaction time 40 min, pH 11.0, dose 5.0 mg/10 mL and temperature 298 K were attuned. The upshot of this process is given in Figure 2, that evidently displays the sorption of 5.0, 10.0, 13.50, 16.0, 16.0 and 16.0 µg/mg at 25.0, 50.0, 75.0, 100.0, 125.0 and 150.0 µg/L concentrations for amino-phenol. Whereas in the event of phenol sorption of 5.0, 10.0, 14.0, 17.0, 17.0 and 17.0 µg/mg at 25.0, 50.0, 75.0, 100.0, 125.0 and 150.0 µg/L concentrations were gotten. It obviously confirms that an extra increase in concentration from 100.0–150.0 µg/L could not boost the sorption. So, a typical of concentration of 100 µg L^−1^ was preferred for the remaining experiments.

#### 2.4.2. Extraction Time

With an augment of interaction time from 5.0 to 60.0 min, the sorption frequently gets augmented for aminophenol and phenol till 40.0 min. Then the sorption remained static for both the molecules. Limitations such as concentration, pH, dose and temperature were 100 µg L^−1^, 11.0, 5.0 mg/10 mL and 298 K were utilized, correspondingly. The contact time effect on sorption is revealed in Figure 3. It may be observed from the figure the sorption in aminophenol and phenol were 4.0, 6.0, 10.0, 14.0 and 16.0 µg/mg that is, 20.0, 30.0, 50.0, 70.0 and 80.0% at 5.0, 10.0, 20.0, 30.0 and 40.0 min and 4.0, 6, 10.50, 14.50 and 17 µg/mg that is, 20.0, 30.0, 50.0, 72.50 and 85.0% at 5.0, 10.0, 20.0, 30.0 and 40.0 min, correspondingly. Additionally, with a rise in contact time the sorption remained unaltered for both molecules. So, at 40.0 min contact time, the rest runs were done.

#### 2.4.3. pH of the Solutions of Phenols

In the event of pH effect on the sorption, primarily the sorption of aminophenol and phenol increased from 2.0 to 16.0 µg/mg, that is, 10.0–80.0% at 9.0–10.50 pH; followed by the unaffected until 13.0. The extra limitations used were contact time (40.0 min), concentration (100 µg L^−1^), dose (5.0 mg/10 mL) and temperature (298 K). The effect of pH on sorption is revealed in Figure 4. A review of this figure shows that the quantity of sorption for aminophenol was 2.0, 3.0, 5.0, 16.0 and 16.0 µg/mg, that is, 10.0, 15.0, 25.0, 80.0 and 80.0% at 9.0, 9.50, 10.0, 10.50 and 11.0 pH. Whilst for phenol at similar pH values, the attained sorption percentages were 10.0, 15.0, 25.0, 80.0 and 80.0. Also, for both molecules the further alteration in pH could not augment the sorption. So, a pH of 11.0 remained a standard one. This performance of phenol sorption was because of pKa values of phenol and *p*-amino-phenol (9.95 and 10.30).

#### 2.4.4. Amount of Nanosorbent Enclosed in the Membrane Cone

The effect of the amount of nanosorbent enclosed in the membrane cone was observed while making the other limitations at contact time (40.0 min), pH (11.0), concentration (100 µg/L) and temperature (298 K). The amount of the nanosorbent effect enclosed in the membrane cone on the sorption of phenols is revealed in Figure 5. It was seen that up to 5.0 mg/10 mL, the quantity of sorption gradually augmented and then it remained unchanged. At doses of 1.0, 2.0, 3.0, 4.0, 5.0, 6.0 and 7.0 mg/10.0 mL the removal of 3.0, 7.0, 10.0, 13.0, 16.0, 16.30, 16.50 µg/mg with 15.0, 35.0, 50.0, 65.0, 80.0, 81.50 and 81.50%, correspondingly, was noted in case of aminophenol. Whilst a similar quantity of uptake was 3.0, 7.0, 10.50, 13.50, 17.0, 17.50 and 17.60 µg/mg with 15.0, 35.0, 52.50, 67.50, 85.0, 87.0 and 87.0% for phenol. So, 5.0 mg/mL was the best quantity of nanosorbent enclosed in the membrane cone.

#### 2.4.5. Desorption Time

While keeping the parameters such as the 40 min contact time, 100 µg/L concentration, 5.0 mg/10 mL dose and 298 K temperature, the effect of time on desorption for aminophenol and phenol was experimented and it was found that with an increase in time interval from 5 to 60 min, % recoveries of both increased constantly (Figure 6) but, on reaching 60 min, % recoveries remained unaffected that is, constant % recoveries. From Figure 3, it is clear at 5, 10, 20, 30, 40 and 50 min that the % recoveries for aminophenol were 42, 58, 65, 75, 85 and 100 and for phenol 40, 55, 60, 70, 80 and 100, respectively. Therefore, it is concluded that, with further increase in time interval after 60 min, the desorption could not be changed.

#### 2.4.6. Desorption pH

Another experiment in which the effect of pH on the desorption of aminophenol and phenol was tested and other limitations used were a 40 min contact time, 100 µg/L concentration, 5.0 mg/10 mL dose and 298 K temperature. The readings showed that when pH was increased from 9 to 12 for aminophenol and phenol the rate of desorption decreased progressively. The effect of pH on desorption is shown in Figure 7 which depicts, at 9, 9.5, 10, 10.5, 11, 11.5 and 12 pH, that the percentage recovery for aminophenol and phenol was 100, 100, 89, 78, 76, 65 and 45 and 100, 100, 90, 77, 75, 60 and 40 respectively. This performance of phenol sorption was because of the pKa values of phenol and *p*-amino-phenol (9.95 and 10.30).

## 3. Materials and Methods

### 3.1. Chemicals and Reagents

Phenol and 3-aminophenol (Figure 8) and nitro-phenol (internal standard) were obtained from Aldrich Chemical Co., St. Louis, MO, USA. Sodium phosphate and sodium dihydrogen phosphate were supplied by SRL, Mumbai, India. *N*-methyl butyl imidazolium bromide, that is, ionic liquid (IL) was obtained from Fluka, Mumbai, India. Ferrous sulphate and poly vinyl alcohol (PVA) were supplied by Qualigens Mumbai, India. Millipore water was collected by a Millipore-Q system made by Bedford, MA, USA. The polypropylene membrane (0.01 µm) was provided by GVS Filtration Technology, Italy. Phosphate buffer (15 mM (pH 7.0)) was arranged utilizing standard protocol. The solutions (10.0–100.0 µg/L) of separable and combination of phenols were arranged in 15.0 milli mole phosphate buffer (pH 7.0) for capillary electrophoresis experiments. The solutions of 0.25–1.5 mg/L were prepared in Millipore water for sample preparation studies.

### 3.2. Instruments Used

The capillary electrophoresis machine was a Quanta 4000 of Waters Chromatography, Millipore, Milord, MA, USA. The software was Millennium 2000 with a data station. The analysis was performed on a fused silica capillary (0.6 m × 75 μm I.D.) and was supplied by Waters, Milord, MA, USA. The pH was adjusted by a pH meter (611, Orion Research Inc., Jacksonville, FL, USA).

### 3.3. Green Synthesis of Nanomaterial

The nanomaterial, that is, composite iron nanoparticles were manufactured utilizing green knowledge as defined elsewhere [32,33,34,35]. Black tea extract was mixed with ferrous sulphate solution and nanoparticles were prepared by green technology. Further, these nanoparticles were allowed to react with ionic liquids to make nanocomposite materials. The instinctive iron nanoparticles were obtained by mingling black tea extract and ferrous sulphate solution. The composite iron nanoparticles were manufactured utilizing *N*-methyl butyl imidazolium bromide ionic liquid.

### 3.4. Fabrication of SPMMTE

The SPMMTE unit was laboratory made as shown in Figure 9. Poly propylene membrane was molded into a cone shape [35 mm × 35 m × 25 m]. The synthesized nanocomposite (5.0 mg) was inserted into a cone shaped poly propylene membrane. This cone was fixed inside 200.0 µL pipette micro tip. The micro tip was dipped in acetone for about 15 min for fixing.

### 3.5. Capillary Electrophoretic Conditions

The capillary electrophoresis machine is described as above. The UV detector (214 nm) was on the cathode side of the machine. The background electrolyte utilized was phosphate buffer (pH 7.0, 15 mM). The experiments were made at 18 kV used voltage. The samples were laden for 30 s via hydrostatic manner of injection. The data were collected at 20 points/second. The experiments were conducted at 23 ± 1 °C temperature. The identification of the separated phenols in the sample was done by comparing the electropherograms of standard ones. The migration times of single phenols were matched with the migration times of the phenols mixture. The quantitative analyses of the phenols in water samples was ascertained using the peak areas of the standard phenols. The internal standard (nitro-phenol) was added before and quantification was achieved by using ratios of peak height or area of the component to the internal standard.
Conc._unknown_ = [AISK/AISU] × [AU/AK] × [Conc._known_],(1)
where AISK = area of peak of internal standard in known, AISU = area of peak of internal standard in unknown, AU = area of unknown, AK = area of known.

### 3.6. Extraction of Phenols from Wastewater by SPMMTE

The solution (0.25–1.5 mg/mL) of the two phenols mixtures were made in pure water. Wastewater samples were collected from the mess pipe line and filtered via Whatman filter paper No. 24 (pre-saturated with phenols). The solutions of the phenol mixture, that is, 1.0 mL of 0.25–1.5 mg/L concentrations were mixed to 9.0 mL water samples to get concentrations from 25–150 µg/L. The wastewater sample was shaken for about 60 min and kept at room temperature overnight. The extraction of phenols from wastewater samples was carried out using the SPMMTE method.

The SPMMTE tip was dipped in the wastewater sample (10 mL) with constant stimulation with a magnetic stirrer for 30 min. A 200 µL sample was engulfed in the SPMMTE tip at a gap of 15 s. The engulfed wastewater was freed back into the beaker. This exercise was performed for 40 min. The cone shaped membrane was taken away from SPMMTE and air dried. This cone shaped structure was shaken with methanol (25 mL) to extract the phenols. The methanol solution was condensed to 0.5 mL in a vacuum. The resultant solution was utilized for determining the concentrations of the phenols using capillary electrophoresis.

## 4. Conclusions

From the results and discussion, it is clear that the combination of solid phase micro membrane tip extraction and capillary electrophoresis is ideal for the monitoring of toxic phenol and *p*-amino-phenol in wastewater. The detection limits of phenol and *p*-amino-phenols were 0.1 and 0.2 µg/L after extraction and CE. The percentage recoveries are quite satisfactory for both the phenols. The method is also fast as it can be finished within 12 min. The optimized circumstances were 100 g/L concentration, 40 min contact time, 11 pH, 5 mg/mL nanoparticles amounts, 60 min desorption time, 9 desorption pH and 298 K temperature. The maximum percent uptakes of *p*-amino-phenol and phenol were 80.0 and 85.0%. The maximum percentage recoveries of *p*-amino-phenol and phenol from the nanomaterial were 99.0 and 98 %. Consequently, the actual extractions of *p*-amino-phenol and phenol from wastewater were 79.2 and 83.30 percent. Therefore, this combination of solid phase micro membrane tip extraction and capillary electrophoresis may be considered as suitable for the monitoring of toxic phenol and *p*-amino-phenol in any water sample.

## Figures and Tables

**Figure 1 molecules-24-03443-f001:**
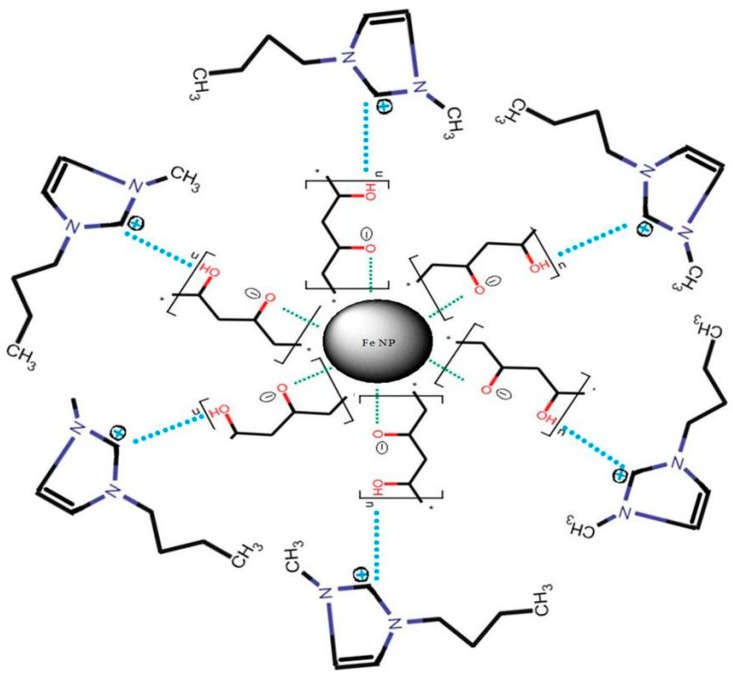
The supra molecular structure of the composite iron nanoparticle.

**Figure 2 molecules-24-03443-f002:**
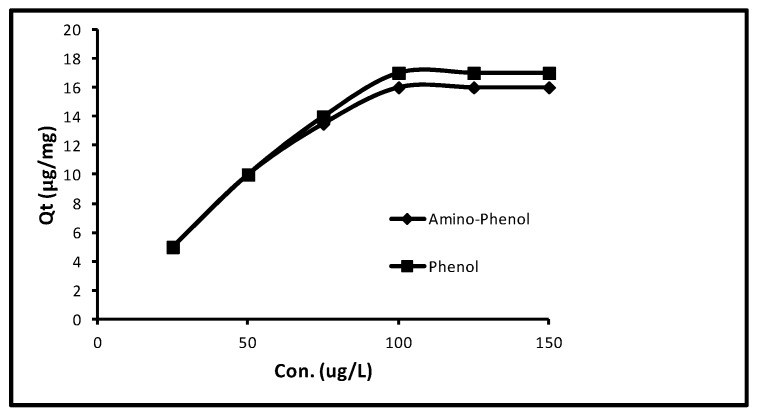
Effect of initial concentrations of phenols.

**Figure 3 molecules-24-03443-f003:**
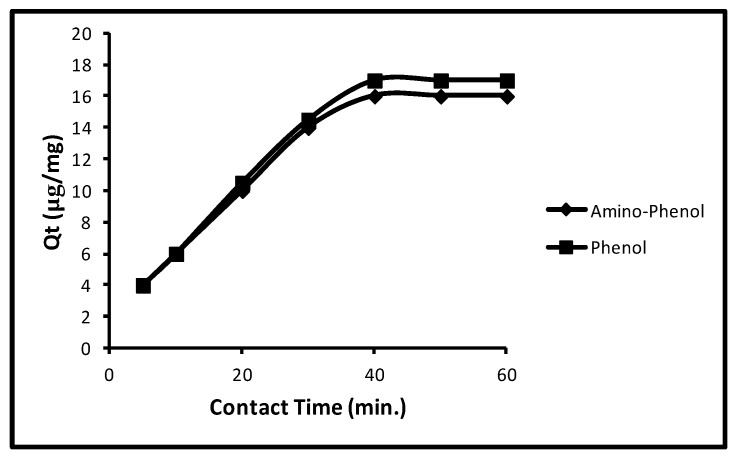
Effect of extraction time of phenols.

**Figure 4 molecules-24-03443-f004:**
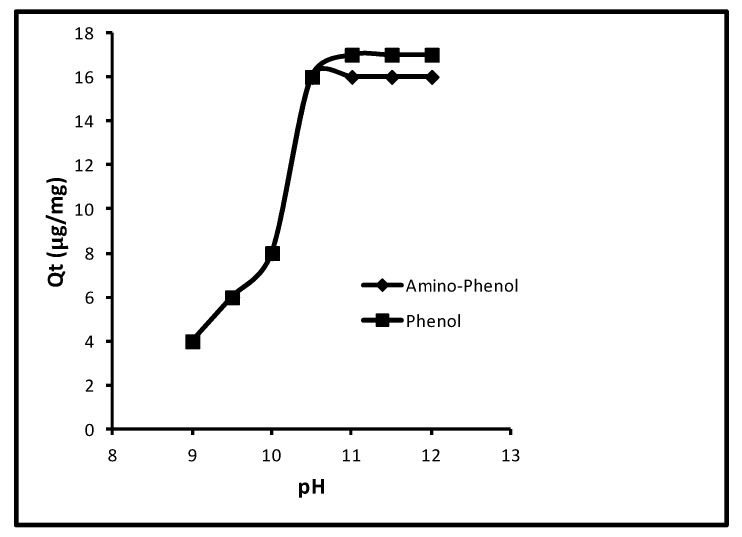
Effect of pH of the solutions of phenols.

**Figure 5 molecules-24-03443-f005:**
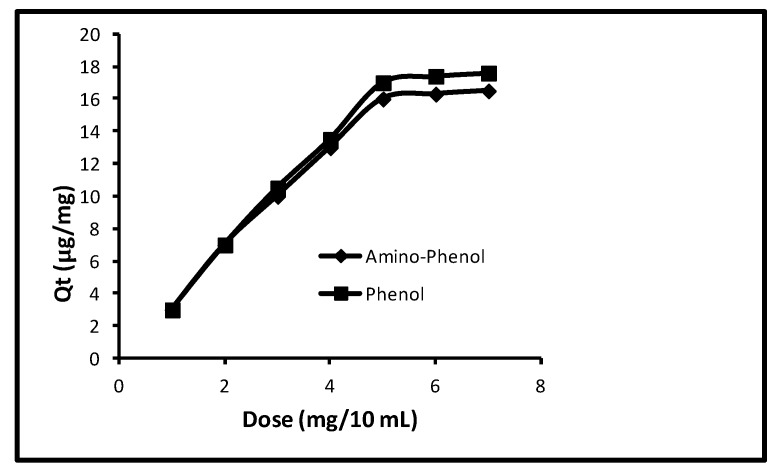
Effect of amount of nanoparticles enclosed in the membrane cone.

**Figure 6 molecules-24-03443-f006:**
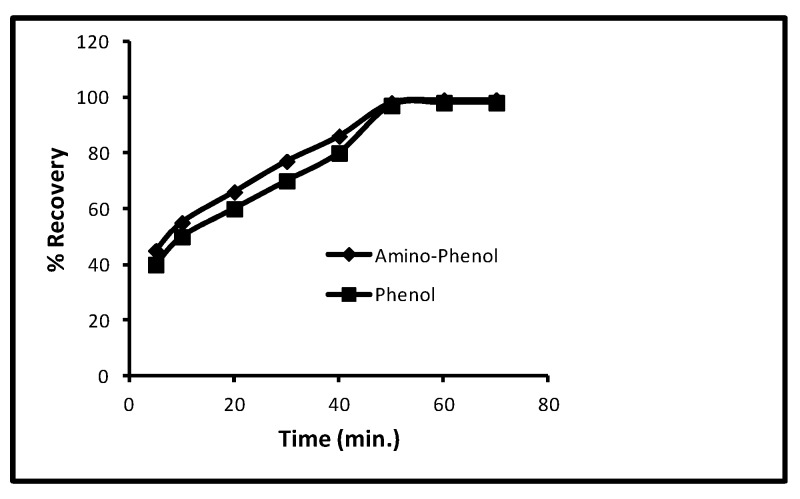
Effect of desorption time for phenols.

**Figure 7 molecules-24-03443-f007:**
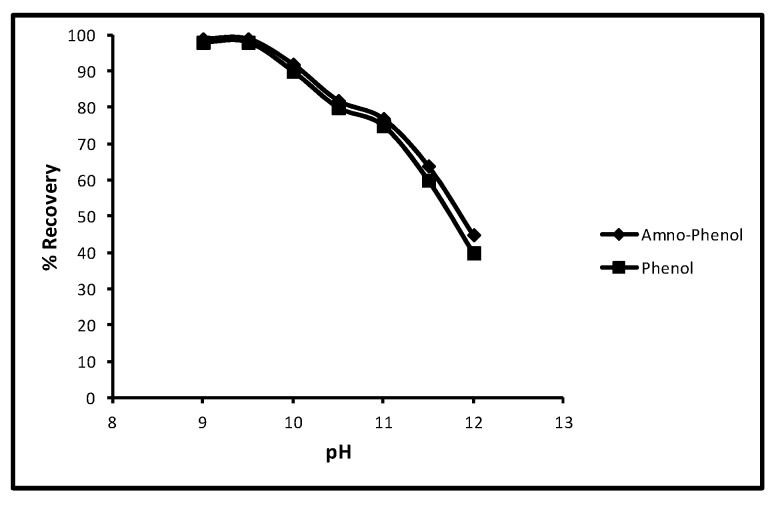
Effect of pH for desorption of phenols.

**Figure 8 molecules-24-03443-f008:**
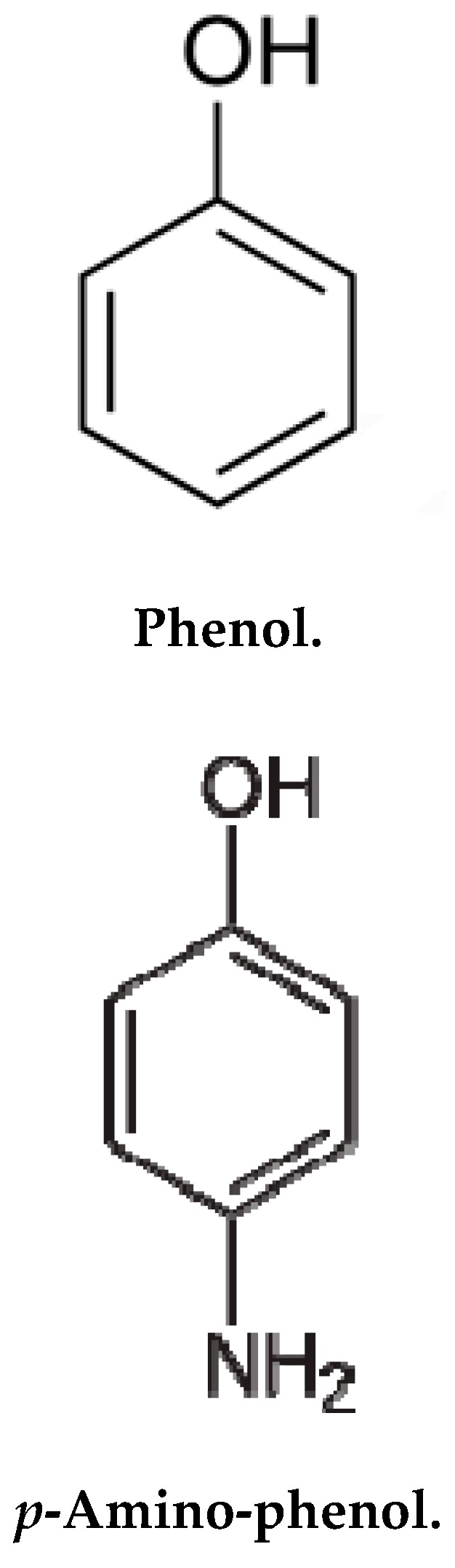
Chemical structures of phenol and *p*-nitro-phenol.

**Figure 9 molecules-24-03443-f009:**
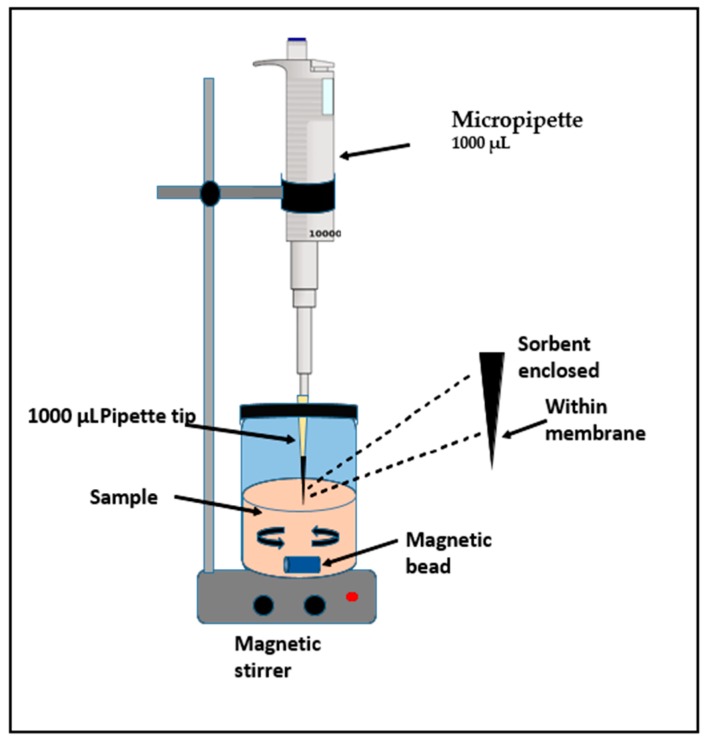
Solid phase micro membrane tip extraction assembly.

**Table 1 molecules-24-03443-t001:** Validation parameters of capillary electrophoresis.

Sl. No.	Validated Parameters	% RSD	Correlation Coefficients	Confidence Levels
1.	Linearity	0.55–1.20	0.9996–0.9997	97.0–96.2
2.	LOD	0.85–1.30	0.9995–0.9996	95.5–96.1
3.	LOQ	0.83–0.102	0.9995–0.9996	96.6–96.0
4.	Specificity	0.73–0.93	0.9995–0.9996	97.0–96.1
5.	Accuracy	0.69–0.87	0.9996–0.9998	96.2–96.2
5.	Precision	0.58–0.71	0.9996–0.9998	96.2–97.0
6.	Ruggedness	0.89–1.40	0.9995–0.9996	96.7–96.6

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
