# Peer review of "Application of Composite NanoMaterial to Determine Phenols in Wastewater by Solid Phase Micro Membrane Tip Extraction and Capillary Electrophoresis"

_molecules, 2019, doi:10.3390/molecules24193443_

Round 1

Reviewer 1 Report

Line 85: Although reference is cited, I would briefly describe the method here. It then gives readers comprehensive information.

Line 92: figure 2 Adsorbent or sorbent? If you haven’t done kinetic experiments to prove this adsorption, please refer that as sorption or sorbent.

Line 124: It is not enough to say that characterization has been done elsewhere.  Need to provide these data at least as supporting information.

Line 137: Detailed description is needed for best conditions.

Line 142 How did authors calculated LOD and LOQ? Also provide calibration curves, may be as supporting information.

Author Response

Please see the attach.

Reviewer 2 Report

The work concerns the combination of solid-phase microextraction using nanomaterial and capillary electrophoresis for the determination of toxic phenol and p-amino-phenol in wastewater. While the idea itself seems interesting, its realization and description are not suitable for publication in their current form. My reservations and comments below.

Why did the authors choose capillary electrophoresis for quantification of phenol and p-amino-phenol? The method applied is new or the authors found it in literature? I can not see the results of CE at all? 25, 26-The migration times of phenol and p-amino-phenol and were 9.0 and 12.0 minutes. The detection limits of phenol and p-amino-phenols were 0.1 and 0.2 µg/L.-Does these sentences concern electrophoresis or extraction it is not clear. 27 remove “ideal” and substitute by another appropriate word. To justify this statement You need to compare your method with other ones already existed for this purpose. Correct the following sentence: First part of the experiment is becomes important in case of the samples of biological and environmental origins.”-into: The first part of the experiment becomes important in case of the samples of biological and environmental origins. 38-correct: „Many ample..”- into Many sample…. Correct the following sentence because it is obscure: “The poisonousness and tumorigenesis concerned investigators to discover the suitable sample preparation and analysis approaches”. 1 should be deleted because it is too obvious for the chemical journal. The documentation of the separated phenols in the sample was done by experimenting the electropherograms of single phenol under the same circumstances of CE.-It is not clear. 104-The following description and quantification are not professional:” The quantitative analyses of the phenols in water samples was conducted by associating the peak  areas of the single phenols with the peak areas of the phenols in sample.” Authors wrote : Waste water samples were collected from the mess pipe line and filtered via Whatman filter paper No. 24.”-The question is: Did you check the sorption properties of the filter paper towards phenols? From the description of the method of sample preparation, it appears that the standard addition method was used for quantitative assessment and that phenol desorption occurs under the influence of methanol. It is different in the abstract. This should be improved so that it is uniform. Subchapter 3.1 should be deleted because the synthesized composite iron nanoparticles has been already described previously ref.22-26. Identification of phenols was done by noting their migration times. Spectrophotometric identification by matching spectra is a much better choice. “The documentation of the separated phenols was carried out using standards of the phenols.”-What does it mean? 139 Specify what parameter applies to the value of the presented correlation coefficient? Table 1 is not clear for instance what is the correlation coefficient for LOD or LOQ? There is no mathematical description of sorption isotherms. The results obtained with the method described should be compared with the methods already described in the literature. None of the graphs has points with error bars. How many times the experiment was repeated?

Author Response

Please see the attach.

Round 2

Reviewer 1 Report

Line 219 and 228: Please change adsorbent to sorbent. Please be consistent through the manuscript.

Author Response

please see the attach

Reviewer 2 Report

The paper has been corrected but still requires some improvement:
1. Authors wrote : Waste water samples were collected from the mess pipe line and filtered via Whatman filter paper No. 24.”-The question is: Did you check the sorption properties of the filter paper towards phenols? Reply: Filter papers were socked with phenol water, washed thoroughly with distilled water, dried and then used.
This is not answer for my question.
2. 139 Specify what parameter applies to the value of the presented correlation coefficient? Table 1 is not clear for instance what is the correlation coefficient for LOD or LOQ? There is no mathematical description of sorption isotherms. Reply: The correlation coefficients in Table 1 are associated with linearity, LOD, LOQ, specificity, accuracy, precision and ruggedness.
This is not answer for my whole doubts.
3. The results obtained with the method described should be compared with the methods already described in the literature. None of the graphs has points with error bars. How many times the experiment was repeated? Reply: The method cannot be compared as there is no method available for other phenols using solid phase micro membrane tip extraction.

The question was about comparison with methods already existed according to accuracy, LOD, LOQ, precision or simplicity of the methods.
4. 104-The following description and quantification are not professional:” The quantitative analyses of the phenols in water samples was conducted by associating the peak areas of the single phenols with the peak areas of the phenols in sample.” Reply: It is corrected to make the language clear.
It was not about the language but about appropriate equations that should be given to illustrate your calculation method.
5. Why did the authors choose capillary electrophoresis for quantification of phenol and p-aminophenol? The method applied is new or the authors found it in literature? I can not see the results of CE at all? Reply: Capillary electrophoresis is the most inexpensive method involving low cost 10 mL buffer only. The method used is developed by the authors and already published. Please ref 26 of the manuscript.
I cannot see the results of separation by CE.

Author Response

please see the attach
